META-RESEARCH ARTICLE

# Tapping into non-English-language science for the conservation of global biodiversity

**Tatsuya Amano**[1,2]*, **Violeta Berdejo-Espinola**[1,2], **Alec P. Christie**[3,4], **Kate Willott**[3], **Munemitsu Akasaka**[5,6], **András Báldi**[7], **Anna Berthinussen**[8], **Sandro Bertolino**[9], **Andrew J. Bladon**[3], **Min Chen**[10,11], **Chang-Yong Choi**[12], **Magda Bou Dagher Kharrat**[13], **Luis G. de Oliveira**[14], **Perla Farhat**[13], **Marina Golivets**[15], **Nataly Hidalgo Aranzamendi**[16], **Kerstin Jantke**[17], **Joanna Kajzer-Bonk**[18,19], **M. Çisel Kemahlı Aytekin**[20], **Igor Khorozyan**[21], **Kensuke Kito**[22], **Ko Konno**[23], **Da-Li Lin**[1,24], **Nick Littlewood**[3,25], **Yang Liu**[26], **Yifan Liu**[27], **Matthias-Claudio Loretto**[28,29], **Valentina Marconi**[30,31], **Philip A. Martin**[3,4], **William H. Morgan**[3], **Juan P. Narváez-Gómez**[32,33], **Pablo Jose Negret**[2,34], **Elham Nourani**[28,29], **Jose M. Ochoa Quintero**[35], **Nancy Ockendon**[36], **Rachel Rui Ying Oh**[1,2,37,38], **Silviu O. Petrovan**[3], **Ana C. Piovezan-Borges**[39], **Ingrid L. Pollet**[40], **Danielle L. Ramos**[41], **Ana L. Reboredo Segovia**[42], **A. Nayelli Rivera-Villanueva**[43], **Ricardo Rocha**[3,44,45], **Marie-Morgane Rouyer**[46], **Katherine A. Sainsbury**[3,47], **Richard Schuster**[48], **Dominik Schwab**[49], **Çağan H. Şekercioğlu**[20,50], **Hae-Min Seo**[12], **Gorm Shackelford**[3,4], **Yushin Shinoda**[5], **Rebecca K. Smith**[3], **Shan-dar Tao**[51], **Ming-shan Tsai**[52], **Elizabeth H. M. Tyler**[3], **Flóra Vajna**[7,53], **José Osvaldo Valdebenito**[54,55], **Svetlana Vozykova**[56], **Paweł Waryszak**[57], **Veronica Zamora-Gutierrez**[58], **Rafael D. Zenni**[59], **Wenjun Zhou**[26], **William J. Sutherland**[3,4]

1 School of Biological Sciences, The University of Queensland, Brisbane, Queensland, Australia, 2 Centre for Biodiversity and Conservation Science, The University of Queensland, Brisbane, Queensland, Australia, 3 Conservation Science Group, Department of Zoology, University of Cambridge, Cambridge, United Kingdom, 4 BioRISC, St. Catharine's College, Cambridge, United Kingdom, 5 Institute of Agriculture, Tokyo University of Agriculture and Technology, Fuchu, Tokyo, Japan, 6 Institute of Global Innovation Research, Tokyo University of Agriculture and Technology, Fuchu, Tokyo, Japan, 7 Lendület Ecosystem Services Research Group, Institute of Ecology and Botany, Centre for Ecological Research, Vácrátót, Hungary, 8 Conservation First, Ampleforth, York, United Kingdom, 9 Department of Life Sciences and Systems Biology, University of Turin, Torino, Italy, 10 School of Life Sciences, Institute of Eco-Chongming (IEC), East China Normal University, Shanghai, China, 11 Yangtze Delta Estuarine Wetland Ecosystem Observation and Research Station, Ministry of Education & Shanghai Science and Technology Committee, Shanghai, China, 12 Department of Agriculture, Forestry, and Bioresources, Seoul National University, Seoul, Republic of Korea, 13 Laboratoire Biodiversité et Génomique Fonctionnelle, Faculté des Sciences, Université Saint-Joseph, Campus Sciences et Technologies, Beirut, Lebanon, 14 Joint Nature Conservation Committee, Peterborough, United Kingdom, 15 Department of Community Ecology, Helmholtz Centre for Environmental Research–UFZ, Halle, Germany, 16 School of Biological Sciences, Monash University, Melbourne, Victoria, Australia, 17 Center for Earth System Research and Sustainability, University of Hamburg, Hamburg, Germany, 18 Institute of Nature Conservation, Polish Academy of Sciences, Kraków, Poland, 19 Department of Invertebrate Evolution, Jagiellonian University, Kraków, Poland, 20 Department of Molecular Biology and Genetics, Koç University, Rumelifeneri Yolu Sarıyer, Istanbul, Turkey, 21 Department of Conservation Biology, Georg-August-Universität Göttingen, Göttingen, Germany, 22 Department of Ecosystem Studies, Graduate School of Agricultural and Life Sciences, The University of Tokyo, Tokyo, Japan, 23 School of Natural Sciences, Bangor University, Gwynedd, United Kingdom, 24 Endemic Species Research Institute, Jiji, Nantou, Taiwan, 25 Department of Rural Land Use, SRUC, Aberdeen, United Kingdom, 26 State Key Laboratory of Biocontrol, School of Ecology, Sun Yat-sen University, Guangzhou, Guangdong, China, 27 School of Agriculture and Biology, Shanghai Jiao Tong University, Shanghai, China, 28 Department of Migration, Max Planck Institute of Animal Behavior, Radolfzell, Germany, 29 Department of Biology, University of Konstanz, Konstanz, Germany, 30 Faculty of Natural Sciences, Department of Life Sciences (Silwood Park), Imperial College London, Ascot, Berkshire, United Kingdom, 31 Institute of Zoology, Zoological Society of London, London, United Kingdom, 32 Departamento de Botânica, Instituto de Biociências, Universidade de São Paulo, Cidade Universitária, São Paulo, Brasil, 33 Forest Ecology and Conservation Group, Conservation Research Institute and Department of Plant Sciences, University of Cambridge, Cambridge, United Kingdom, 34 School of Earth and Environmental Sciences, The University of Queensland, Queensland, Australia, 35 Instituto de Investigación de Recursos Biológicos Alexander von

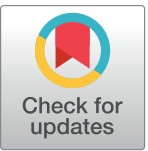

**Data Availability Statement:** All data used in the analysis are available as supplementary files (S1–S5 Data) and all codes used in the analysis are available at: http://doi.org/10.17605/OSF.IO/Y94ZT.

**Funding:** This work was funded by the Australian Research Council Future Fellowship FT180100354, the University of Queensland strategic funding (T. A.), the Natural Environment Research Council NE/ L002507/1 (A.P.C.), National Research, Development and Innovation Office, Hungary ED_18-1-2018-0003 (A. Báldi), University of Turin, local research grant (S.B.), the German Research Foundation under Germany's Excellence Strategy EXC 2037 Project Number 390683824 (K.J.), Grant Sonata Bis 4 no. 2014/14/E/NZ8/00165 from the National Science Centre, Poland (J.K.-B,), German Research Foundation grant 2153/5-1 (I.K.), European Union's Horizon 2020 research and innovation programme under the Marie Skłodowska-Curie grant agreement no. 798091 (M.-C.L.), the Natural Environment Research Council NE/P012345/1 (V.M.), Coordenação de Aperfeiçoamento de Pessoal de Nível Superior – Brasil (CAPES) – Finance Code 001 (J.P.N.G.), Colombian Administrative Department of Science, Technology and Innovation (P.J.N.), Consejo Nacional de Ciencia y Tecnología-CONACYT 1004537 (A.N.R.V.), European Union's Horizon 2020 research and innovation programme under the Marie Skłodowska-Curie grant agreement no. 766417 (M.-M.R.), Ph.D. fellowship at the University of Veterinary Medicine Budapest (F.V.), Chilean National Agency for Research and Development, BECAS CHILE 72170569 (J.O.V.), CNPq-Brazil 304701/2019-0 (R.D.Z.), Arcadia, MAVA, and the David and Claudia Harding Foundation (W.J.S.). The funders had no role in study design, data collection and analysis, decision to publish, or preparation of the manuscript.

**Competing interests:** The authors have declared that no competing interests exist.

**Abbreviations:** BA, Before–After; BACI, Before– After–Control–Impact; CAR, conditional autoregressive; CI, Control–Impact; GLM, generalised linear model; GLMM, generalised linear mixed model; IUCN, International Union for Conservation of Nature; MCMC, Markov chain Monte Carlo; RCT, Randomised Controlled Trial.

Humboldt, Bogotá, Colombia, **36** Endangered Landscapes Programme, The Cambridge Conservation Initiative, Cambridge, United Kingdom, **37** German Centre for Integrative Biodiversity Research (iDiv) Halle-Jena-Leipzig, Leipzig, Germany, **38** Helmholtz Centre for Environmental Research (UFZ), Leipzig, Germany, **39** Programa de Pós-Graduação em Ecologia e Conservação, Instituto de Biociências (INBIO), Universidade Federal de Mato Grosso do Sul (UFMS), Campo Grande, Mato Grosso do Sul, Brazil, **40** Department of Biology, Acadia University, Wolfville, Canada, **41** Plantem—Plant Technology and Environmental Monitoring Ltd., Sao Jose dos Campos, Sao Paulo, Brazil, **42** Department of Earth and Environment, Boston University, Boston, Massachusetts, United States of America, **43** Centro Interdisciplinario de Investigación para el Desarrollo Integral Regional Unidad Durango (CIIDIR), Instituto Politécnico Nacional, Durango, México, **44** CIBIO-InBIO, Research Center in Biodiversity and Genetic Resources, University of Porto, Vairão, Portugal, **45** CIBIO-InBIO, Research Center in Biodiversity and Genetic Resources, Institute of Agronomy, University of Lisbon, Lisbon, Portugal, **46** CEFE, Univ Montpellier, CNRS, EPHE, IRD, Montpellier, France, **47** Faculty of Kinesiology, Sport, and Recreation, University of Alberta, Edmonton, Alberta, Canada, **48** Department of Biology, Carleton University, Ottawa, Ontario, Canada, **49** Agroecology, Department of Crop Sciences, University of Göttingen, Göttingen, Germany, **50** School of Biological Sciences, University of Utah, Salt Lake City, Utah, United States of America, **51** Graduate School of Engineering and Science, University of the Ryukyus, Senbaru, Nishihara, Nakagami, Okinawa, Japan, **52** Wildlife Conservation Research Unit, Department of Zoology, University of Oxford, Oxford, United Kingdom, **53** Department of Ecology, Institute for Biology, University of Veterinary Medicine, Budapest, Hungary, **54** Milner Centre for Evolution, University of Bath, Bath, United Kingdom, **55** Department of Evolutionary Zoology and Human Biology, University of Debrecen, Hungary, **56** Faculty of Energy and Ecotechnology (GreenTech), ITMO University, St Petersburg, Russia, **57** School of Life and Environmental Sciences, Centre for Integrative Ecology, Deakin University, Victoria, Australia, **58** Cátedras CONACYT—Centro Interdisciplinario de Investigación para el Desarrollo Integral Regional Unidad Durango (CIIDIR), Instituto Politécnico Nacional, Durango, México, **59** Departamento de Ecologia e Conservação, Instituto de Ciências Naturais, Universidade Federal de Lavras, Campus Universitário, Lavras, Minas Gerais, Brazil

\* t.amano@uq.edu.au

## Abstract

The widely held assumption that any important scientific information would be available in English underlies the underuse of non-English-language science across disciplines. However, non-English-language science is expected to bring unique and valuable scientific information, especially in disciplines where the evidence is patchy, and for emergent issues where synthesising available evidence is an urgent challenge. Yet such contribution of non-English-language science to scientific communities and the application of science is rarely quantified. Here, we show that non-English-language studies provide crucial evidence for informing global biodiversity conservation. By screening 419,679 peer-reviewed papers in 16 languages, we identified 1,234 non-English-language studies providing evidence on the effectiveness of biodiversity conservation interventions, compared to 4,412 English-language studies identified with the same criteria. Relevant non-English-language studies are being published at an increasing rate in 6 out of the 12 languages where there were a sufficient number of relevant studies. Incorporating non-English-language studies can expand the geographical coverage (i.e., the number of 2° × 2° grid cells with relevant studies) of English-language evidence by 12% to 25%, especially in biodiverse regions, and taxonomic coverage (i.e., the number of species covered by the relevant studies) by 5% to 32%, although they do tend to be based on less robust study designs. Our results show that synthesising non-English-language studies is key to overcoming the widespread lack of local, context-dependent evidence and facilitating evidence-based conservation globally. We urge wider disciplines to rigorously reassess the untapped potential of non-English-language science in informing decisions to address other global challenges.

Please see the Supporting information files for Alternative Language Abstracts.

## Introduction

History demonstrates that important scientific information is published not just in English but also in other languages. The structure of the Nobel Prize–winning antimalarial drug was first published in simplified Chinese [1]. An important rule regarding biodiversity was founded on evidence published in Spanish [2]. Many of the earliest papers on COVID-19 were written, again, in simplified Chinese [3]. Yet the contribution of such non-English-language science to scientific communities, and the broader society, is rarely quantified.

We test this untapped potential of non-English-language science through an assessment of non-English-language studies' contribution to evidence synthesis—the process of compiling and summarising scientific information from a range of sources. Evidence synthesis plays a major role in informing decisions for tackling global challenges in fields such as healthcare [4], international development [5], and biodiversity conservation [6]. To date, non-English-language studies have largely been ignored in evidence synthesis [7–9]. The consequences of this common practice are, however, rarely investigated in most disciplines apart from healthcare. And even there, the focus has almost exclusively been on how including non-English-language studies might change the statistical results of meta-analyses [10,11] (see Supplementary Text for a review of earlier relevant studies). However, non-English-language studies may also enhance the synthesis of evidence with specific types of scientific information that is not available in English-language studies, especially in disciplines dealing with more geographically and taxonomically diverse targets and phenomena than healthcare [12].

Synthesising non-English-language studies could be an effective avenue for reducing the existing, severe gaps in the geographical and taxonomic coverage of available scientific evidence for biodiversity conservation [13,14]. Compiling evidence on what does or does not work in biodiversity conservation, and informing decisions with robust scientific evidence is critical to halting the ongoing biodiversity crisis [6]. As local and context-dependent evidence is crucially required for conservation-related decision-making [15], the geographical and taxonomic gaps in evidence, especially in biodiverse regions, pose a major challenge to our scientific understanding of the biodiversity crisis and the implementation of evidence-based conservation globally. Non-English-language studies could be particularly important in biodiversity conservation for the following reasons. First, over one-third of scientific documents on biodiversity conservation are published in languages other than English [16]. Second, gaps in globally compiled English-language evidence are often found in areas where English is not widely spoken [13]. Third, important evidence in biodiversity conservation is routinely generated by local practitioners, who often prefer publishing their work in their first language, which, for many, is not English [16].

Here, we adopted the discipline-wide literature search method [17] to screen 419,679 peer-reviewed papers in 326 journals, published in 16 languages (S1 Data), to identify non-English-language studies testing the effectiveness of interventions in biodiversity conservation (see Materials and methods). Combining this dataset with English-language studies identified with the same criteria, stored in the Conservation Evidence database [17], enabled us to assess the contribution of non-English-language studies to evidence synthesis through the testing of the following common perceptions that have rarely been corroborated to date: (i) the amount of relevant scientific evidence that is available only in non-English languages is negligible [18];

(ii) the number of relevant studies being published in non-English languages has been decreasing over time [19]; (iii) the quality of non-English-language studies (measured using the study designs adopted; see Materials and methods for more detail) is lower than that of English-language studies [7]; and (iv) evidence published in English represents a random subset of evidence published across all languages [12].

## Results

Our search elicited a total of 1,234 eligible non-English-language studies (including 53 studies on amphibians, 247 on birds, and 161 on mammals, which were used for a detailed species-level comparison with English-language studies) testing the effectiveness of conservation interventions, published in 16 languages (S2 and S3 Data). This adds a considerable amount of scientific evidence for biodiversity conservation to the Conservation Evidence database, which now stores 4,412 English-language studies (including 284 studies on amphibians, 1,115 on birds, and 1,154 on mammals). The proportion of eligible studies in each journal varied among languages, with Japanese (the highest proportion of eligible studies in a journal was 26.7%), Hungarian (15.3%), French (12.9%), and German (9.1%) showing particularly high proportions (largely <5% of the studies screened were eligible in journals of other languages) (S1 Fig). In all languages, except Hungarian, many journals searched had almost no eligible studies, showing that our search had covered and gone beyond most of the relevant journals (see Limitations in Materials and methods for more details).

The yearly number of eligible non-English-language studies published in each journal has increased significantly over time, especially since 2000, in 6 out of the 12 languages covered (French, German, Japanese, Portuguese, Russian, and simplified Chinese), with Portuguese and Russian showing a particularly rapid increase, while traditional Chinese also showed a marginally significant increase (Fig 1). The other 5 languages did not show a significant change in the number of eligible studies over time. This result thus refutes the common perception that the number of non-English-language studies providing evidence is declining. The recent increase in eligible studies indicates that performing searches only on volumes from the most recent 10 years in some long-running journals had minimal impact.

Our results largely support one of the common perceptions—that non-English-language studies tend to be based on less robust study designs. Studies in 10 out of the 16 languages we searched were significantly more likely to adopt less robust designs, compared to English-language studies, when controlling for the effect of study taxa and countries where English-language studies were conducted (Fig 2 and S1 Table). Of the other 6 languages showing no significant difference in designs from English-language studies (Persian, Portuguese, Spanish, traditional Chinese, Turkish, and Ukrainian), only Portuguese and Spanish had reasonable sample sizes to allow a reliable estimation of the proportion of studies adopting different study designs (i.e., 10 or more studies in each taxonomic group), indicating that designs adopted in studies in those 2 languages were comparable to those in English-language studies.

There was a clear bias in study locations between languages. English-language studies were conducted in a total of 952 of the 2° × 2° grid cells and non-English-language studies in 353 grid cells, 238 of which had no English-language studies (those grid cells shown in black in Fig 3; also see Comparing study locations in Materials and methods). Therefore, non-English-language studies expanded the geographical coverage of English-language studies by 25%, though this percentage could be higher when using finer-resolution grid cells. More non-English-language studies tended to be found in grid cells with fewer English-language studies, especially in East/Central/Western Asia, Russia, northern Africa, and Latin America (Figs 3 and S2), but the relationship was not significant when controlling for spatial autocorrelation (posterior

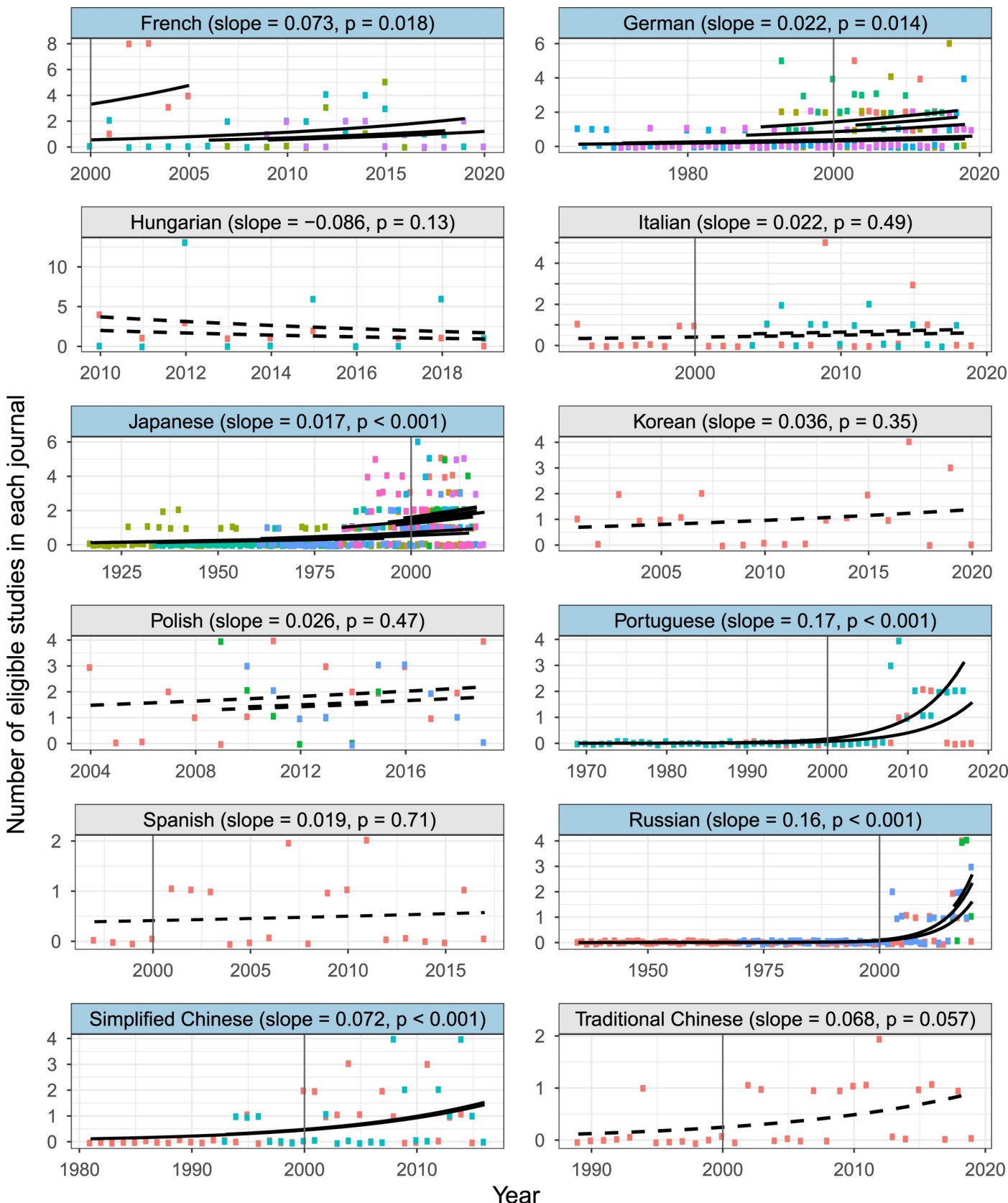

**Fig 1. Language-specific yearly changes in the number of non-English-language studies testing the effectiveness of conservation interventions published in each journal.** Only journals with 10 or more eligible studies are shown (colours indicate different journals), and, thus, 4 languages for which there were no such journals are omitted. Black lines represent regression lines for each journal (solid lines: significant slopes, dashed lines: nonsignificant slopes) based on Poisson generalised linear models with journals as a fixed factor. Languages with a statistically significant positive slope are shown with blue background. Vertical lines indicate the year 2000. This figure was created using S1 and S2 Data with Code 1.

median slope in a conditional autoregressive model: −0.012, 95% credible interval (CI): −0.032 to 0.005; see an inset in Fig 3). Non-English-language studies expanded the geographical coverage based on English-language studies by 12% for amphibians (S3 Fig), 16% for birds (S4

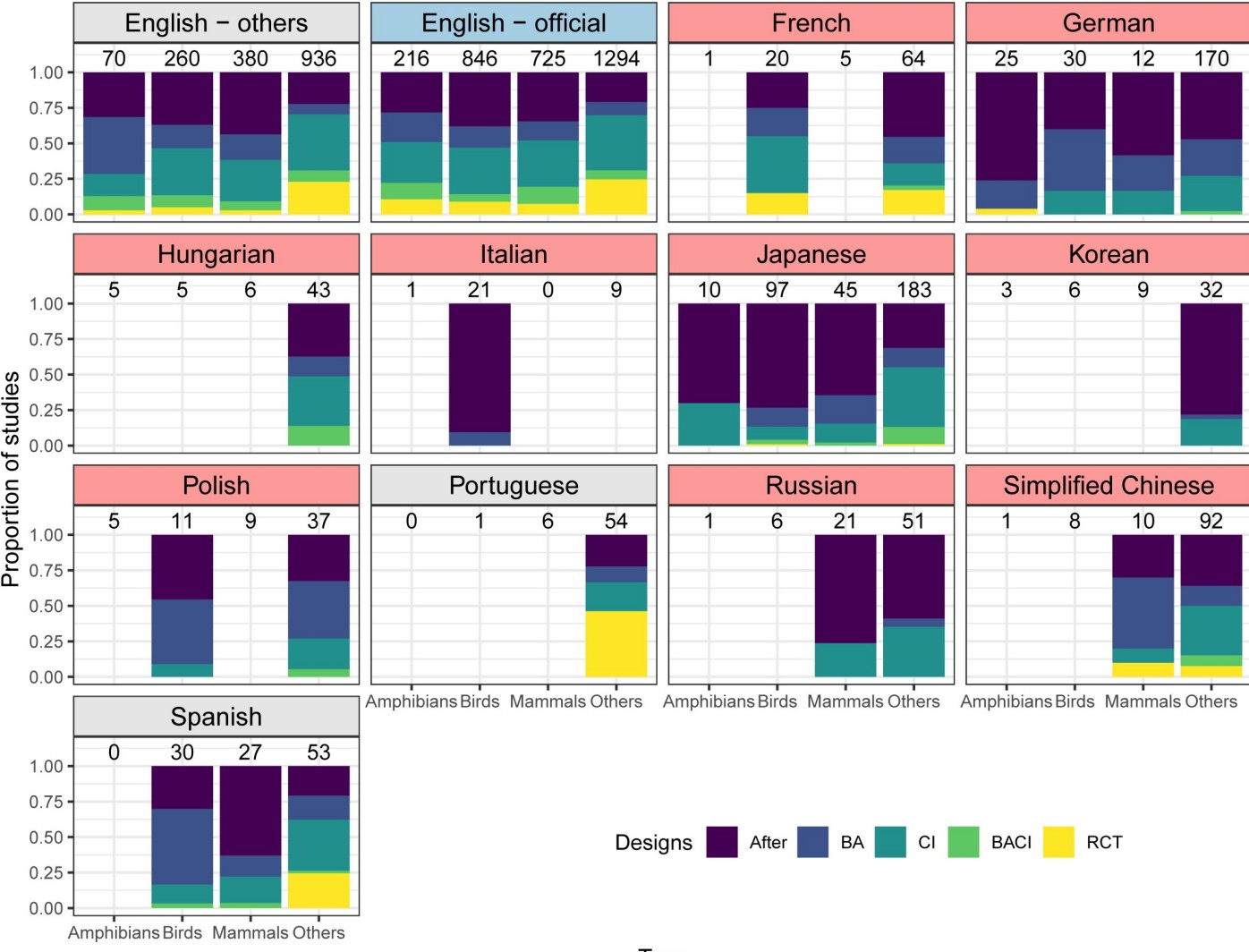

**Fig 2. The proportion of studies in different languages that tested the effectiveness of conservation interventions with different study designs.** Designs in the order of increasing robustness: After, BA, CI, BACI, or RCT. "English–others": English-language studies conducted in countries where English is not an official language. "English–official": English-language studies conducted in countries where English is an official language. Languages with statistically less robust designs compared to "English–others" are shown with pink background, those with statistically more robust designs with blue background, and those with a nonsignificant difference with grey background. The numbers above bars represent the number of studies in each taxon (i.e., amphibians, birds, mammals or others)—language group. Only groups with at least 10 studies are shown. Studies in 5 languages (Arabic, Persian, traditional Chinese, Turkish, and Ukrainian) are not shown as no taxon—language group had 10 or more studies; see S3 Data for study designs adopted in those languages. This figure was created using S3 and S4 Data with Code 2. BA, Before–After; BACI, Before–After–Control–Impact; CI, Control–Impact; RCT, Randomised Controlled Trial.

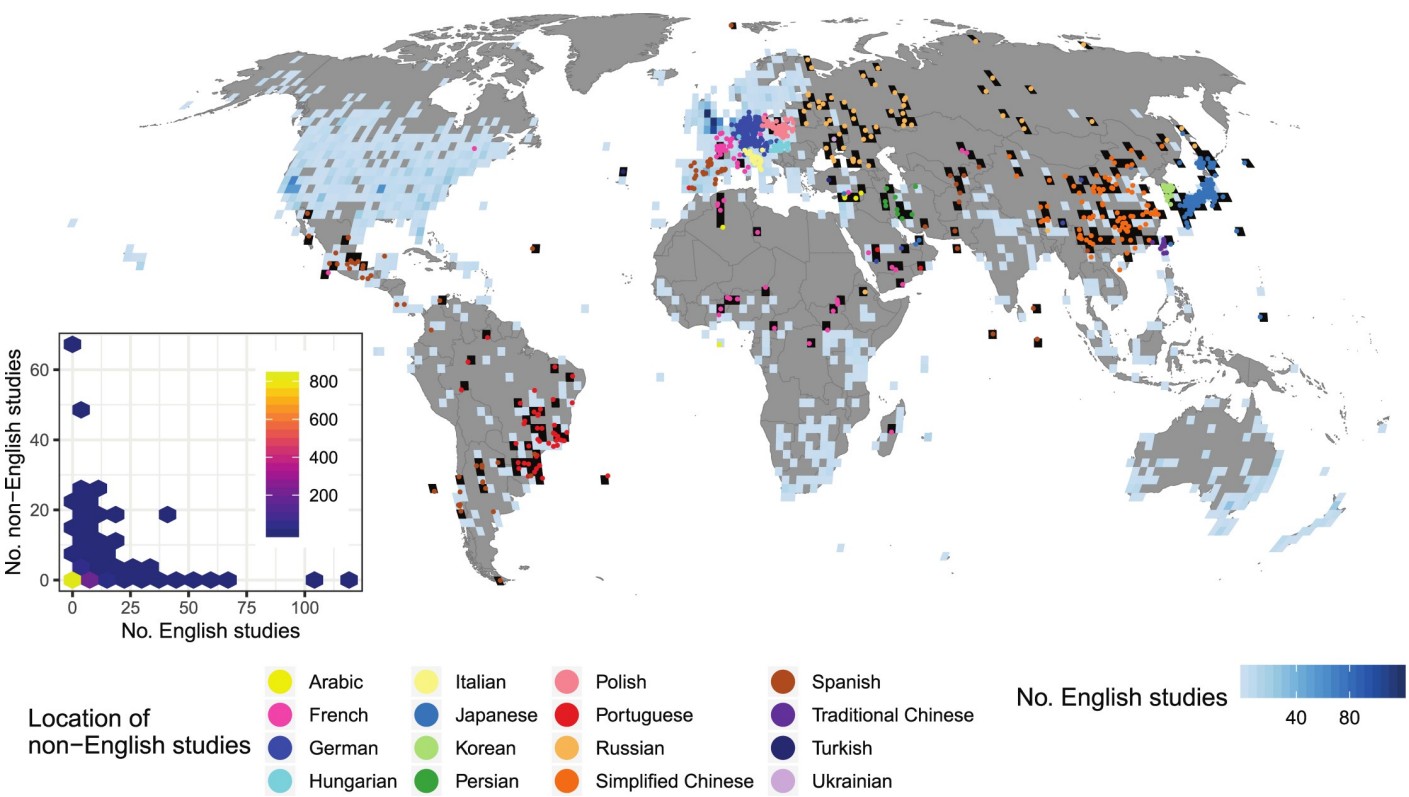

**Fig 3. The location of 1,203 non-English-language studies with coordinate information, compared to the number of English-language studies testing the effectiveness of conservation interventions within each 2° × 2° grid cell (952 grid cells in total).** Non-English-language studies were found in 353 grid cells, 238 of which were without any English-language studies (grid cells in black). The inset is a hexbin chart showing a negative (although nonsignificant) relationship between the number of English-language studies and the number of non-English-language studies (No. non-English studies) within each grid cell. Brighter colours indicate more grid cells in each hexagon. This figure was created using S3 and S4 Data with Code 3. Map produced from the Natural Earth dataset (v.4.1.0) at 1:50 m scale (https://www.naturalearthdata.com/downloads/50m-cultural-vectors/).

Fig), and 12% for mammals (S5 Fig). In all 3 taxa, significantly more non-English-language studies were found in grid cells with fewer English-language studies (amphibians: slope: −0.51, 95% CI: −0.94 to −0.17; birds: slope: −0.23, 95% CI: −0.44 to −0.073; mammals: slope: −0.48, 95% CI: −0.74 to −0.25; also see insets in S3–S5 Figs).

The 1,234 non-English-language studies together provided evidence on the effectiveness of conservation interventions for a total of 1,954 unique species recognised by the International Union for Conservation of Nature (IUCN), including 40 (6 threatened) amphibian, 564 (37 threatened) bird, and 194 (22 threatened and 2 Data Deficient) mammal species. Although species with more studies in non-English languages also tended to have more studies in English for all 3 taxa (generalised linear mixed models for amphibians: slope = 0.12, $z = 7.93$, $p < 0.001$; birds: slope = 0.060, $z = 13.18$, $p < 0.001$; mammals: slope = 0.026, $z = 5.65$, $p < 0.001$; also see insets in Fig 4), non-English-language studies provided scientific evidence on the effectiveness of conservation interventions for an additional 9 amphibian, 217 bird, and 64 mammal species that were not covered by English-language studies (Fig 4), meaning 5%, 32%, and 9% increases in the evidence coverage of amphibian, bird, and mammal species, respectively. Similarly, non-English-language studies increased the evidence coverage of threatened species (Critically Endangered, Endangered, and Vulnerable species classified in the IUCN Red List of Threatened Species) by 23% for birds and 3% for mammals. All threatened amphibian species covered by non-English-language studies were also studied in

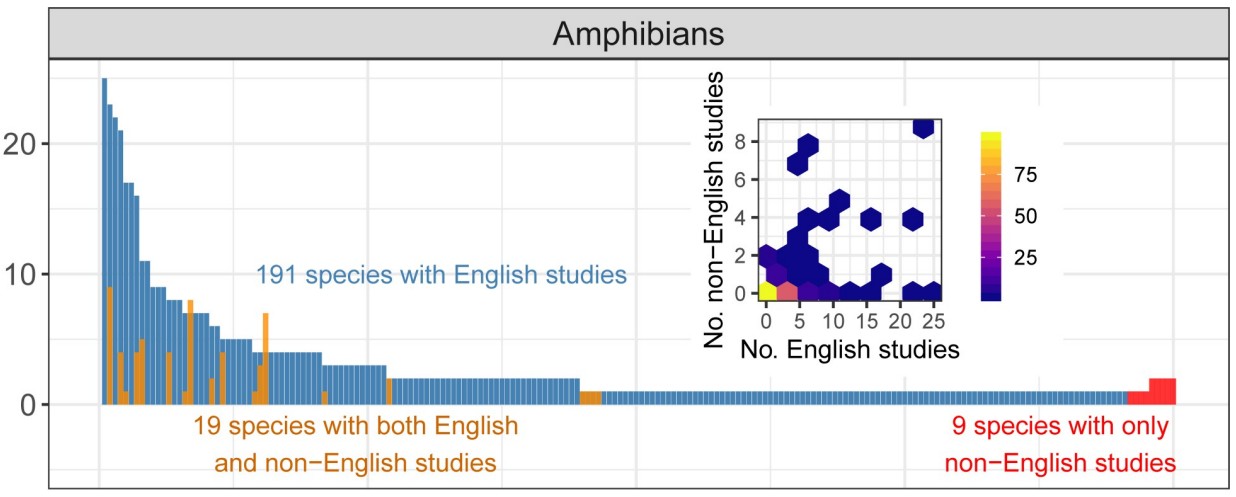

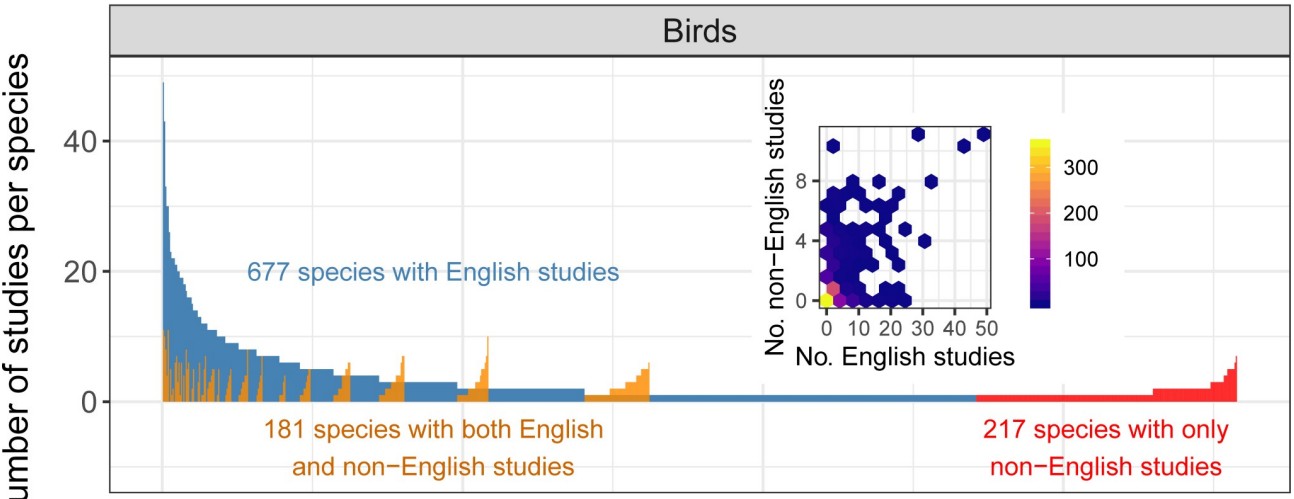

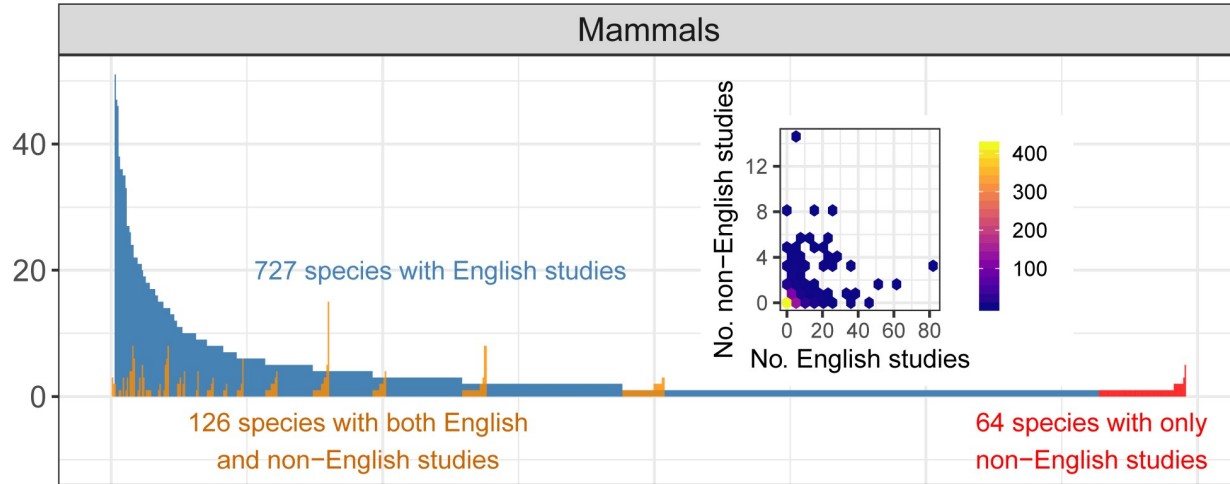

Species in order of decreasing English studies

**Fig 4. The number of English- and non-English-language studies testing the effectiveness of conservation interventions for each amphibian, bird, and mammal species.** The number of English-language studies for each species (blue), with species ranked on the x-axis in order of decreasing English-language studies per species, and the number of non-English-language studies per species for those species studied by both English- and non-English-language studies (orange), and those studied only by non-English-language studies (red). Note that 2 mammal species with 82 and 63 English-language studies are not shown as outliers (see S5 Data). The insets are hexbin charts showing significantly positive relationships between the number of English-language studies (No. English studies) and the number of non-English-language studies (No. non-English studies) per species. Brighter colours indicate more species in each hexagon. Only studies published in 2012 or earlier for amphibians, 2011 or earlier for birds, and 2018 or earlier for mammals were used in this figure. This figure was created using S3 and S5 Data with Code 4.

English-language studies (S6 Fig). Threatened species with more studies in non-English languages had fewer studies in English for birds (slope = −0.34, z = −2.35, *p* = 0.019) but not for mammals (slope = 0.030, z = 0.923, *p* = 0.356; also see insets in S6 Fig. Threatened amphibians could not be modelled as only 2 species were covered by non-English-language studies).

## Discussion

Our analyses demonstrate that 3 out of the 4 common perceptions on the role of non-English-language scientific knowledge are not supported by evidence. We show that, instead, (i) a considerable amount of scientific evidence underpinning effective conservation is available in non-English languages (over 1,000 studies found in our searches); (ii) the number of published studies providing such evidence has been increasing in many languages; and (iii) non-English-language studies can provide evidence that is relevant to species (including threatened species) and locations (including highly biodiverse regions, such as Latin America) for which little or no English-language evidence is available. These results, based on a global empirical analysis of 5,646 studies in 17 languages, corroborate earlier arguments on the potential importance of non-English-language scientific knowledge in evidence-based biodiversity conservation [16]. A poor availability of species- and location-specific evidence, especially in countries where English is not widely spoken, has been recognised as a major impediment to evidence-based conservation [20], as scientific knowledge is often used only if it is relevant to the specific context of policies and practices [15,21]. Meanwhile, a systematic bias in study characteristics, such as species and ecosystems studied, has been found between English- and Japanese-language studies in ecology [12]. Our study attests to the between-language bias in study characteristics, namely study species and locations, at the global scale, showing that incorporating non-English-language studies in evidence syntheses is an effective approach to rectifying biases and filling gaps in the availability of evidence over space and species. Examples of such non-English-language evidence on threatened species include a Spanish-language study testing the use of guardian dogs to alleviate conflicts between low-income livestock farmers in northern Patagonia and carnivores including endangered Andean mountain cats (*Leopardus jacobita*) [22], and a Japanese-language study reporting the effectiveness of relocation for endangered Blakiston's fish owls (*Bubo blakistoni*) [23].

The other perception that non-English-language studies tend to adopt less robust designs seems to be supported by our results, although a reasonable number of non-English-language studies with robust designs also exist, especially in Portuguese (25 studies with Randomised Controlled Trial (RCT)) and Spanish (13 studies with Before–After–Control–Impact (BACI) and 3 with RCT). Scientific evidence presented in non-English-language studies could thus be lower in quality, and suffer from more serious biases, on average, compared to that provided by English-language studies [24]. This difference in evidence quality between English-language and non-English language studies is likely to create a trade-off in evidence-poor regions, between the availability of context-specific evidence and the quality of evidence; for some species and locations, the only available evidence might be found in non-English-language studies

based on less robust designs [25]. Nevertheless, blindly discarding such lower-quality, yet relevant, studies—a common practice in conventional evidence syntheses—could unnecessarily delay, misinform, or hinder evidence-based decision-making, especially in disciplines, such as conservation, where robust evidence bases are patchy [14], and for emergent issues, such as pandemics, where making the best use of available evidence is an urgent challenge [26,27]. A promising approach here is the model-based synthesis of evidence with varying qualities and degrees of relevance to a specific context [24], where non-English-language studies are expected to play a crucial role as an important source of highly context-specific evidence.

We should note, however, that even searching for, and including, non-English-language studies would not fully address the large evidence gaps in some regions faced with the most pressing issues including biodiversity loss, such as Southeast Asia, tropical Africa, and Latin America. Therefore, the use of existing non-English-language science is not a panacea. Generating more local evidence, based on robust study designs, and publishing it in any language should be further encouraged and supported globally, but especially in those evidence-poor regions, for example, through the distribution of free teaching materials to facilitate the testing of conservation interventions [28]. Although not included in this study, non-English-language grey literature could be another potentially important source of evidence [29]; whether and how non-English-language grey literature might fill gaps in English-language evidence still remains to be tested (also see Limitations in Materials and methods for other limitations).

This study showcases the continued vital role of non-English-language studies in providing evidence for tackling the ongoing biodiversity crisis, given the increasing number of relevant studies being published in many non-English languages. However, the degree of importance of such evidence will vary depending on the topic and discipline of focus. Relatively little evidence may be available in non-English languages for a highly specific purpose—for example, for understanding the effectiveness of a single intervention for a specific species—while much evidence may be available for more descriptive purposes, such as for understanding species occurrence. However, for global-scale evidence syntheses with a broad scope, especially those on biodiversity conservation where local studies could play an important role, such as those conducted by the Intergovernmental Science-Policy Platform on Biodiversity and Ecosystem Services, incorporating non-English-language scientific knowledge should become the norm (see Box 1 for our suggestions on how this could be achieved). Generating new scientific knowledge through individual studies requires sizable financial investment, as well as associated time costs [30]. Therefore, making better use of existing knowledge that has yet to be fully utilised due to the language of publication should be a cost- and time-efficient approach for filling gaps and rectifying biases in the evidence base for tackling urgent global challenges. In 1922, the philosopher Ludwig Wittgenstein stated "Die Grenzen meiner Sprache bedeuten die Grenzen meiner Welt" (the limits of my language mean the limits of my world) [31]. Hundred years on, his quote still seems applicable to science today. Scientific communities should stretch the limits of our shared knowledge, and its benefits, by uncovering knowledge that has long been accumulating and continues to be produced in languages other than English.

## Materials and methods

### Searches for non-English-language studies on the effectiveness of conservation interventions

**Objective of the searches.**    The searches aimed to identify peer-reviewed scientific studies (a study is defined as a paper published in a peer-reviewed journal) written in a language other than English that tested the effectiveness of one or more conservation interventions for any species group or habitat. Our search strategy was based on the protocol for discipline-wide

## Box 1. Strategies for and challenges in synthesising non-English-language literature

Searching effectively and understanding non-English-language literature can be a challenging task, with the lack of relevant language skills often being a key reason for excluding non-English-language literature in evidence synthesis [8]. Here, we summarise how we can practically synthesise non-English-language literature under such restrictions.

### How to choose languages

Including more languages would make a synthesis more comprehensive, but given that there are over 7,000 languages in the world, in reality, we would need to choose a set of languages to be covered by each synthesis activity. One option for global synthesis is to cover the 15 languages covered in [16] (chosen based on the national languages of the 20 highest ranked countries in the World Bank's indicator for scientific and technical journal articles [32]). However, what languages are important will vary depending on the topic and region. For example, our results showed that, globally, the proportion of studies testing conservation interventions was especially high in Japanese, Hungarian, French, and German (S1 Fig). A synthesis activity focusing on a certain region/country should obviously cover languages that are widely spoken there, for example, Spanish and Portuguese in Latin America, and Russian in Russia. A useful tool for identifying important languages for bird conservation is an interactive map of bird species richness associated with each of the 120 official languages from each country in the world [33] (https://translatesciences.shinyapps.io/bird_language_diversity/).

### How to develop collaborations

Collaborating with a native speaker of the language of focus is the most effective approach to searching for literature published in a particular language, but finding that collaborator can be a challenge. You could start by asking your colleagues or members of your institute. Recruiting collaborators at conferences or on social media could be effective too. Creating culturally diverse academic environments should make this process easier [34]. Providing your collaborators with appropriate training and guidelines is key to searching literature in a consistent manner. Importantly, in order to develop a healthy and ethical collaboration, you should not exploit the skills and knowledge of the collaborator. Instead, codevelop the project with your collaborators and give them appropriate credit (e.g., in the form of coauthorship) [35].

### Whether to use machine translation

The quality of machine translation has been improving rapidly, especially for languages with sufficient digital resources, such as Spanish, German, Japanese, and French [36], aiding a wider understanding of non-English-language literature [37]. However, even a small number of critical errors, which are still found in machine translation [37], could have major consequences in evidence synthesis, and we still need more robust tests to assess the reliability of machine translation in conducting evidence synthesis. We thus believe that even though qualities are improving rapidly, machine translation should still be used with caution in evidence synthesis, for example, when a native speaker of the language can be asked to double-check the translation output.

## Where to search for non-English-language literature

Many non-English-language journals are not indexed in well-known literature search systems, such as Web of Science and Scopus, although Web of Science incorporates several regional databases that allow searches for non-English-language literature (depending on institutional subscription). Google Scholar is instead an effective approach to searching for non-English-language literature; searches can also be restricted to websites written in the language of focus. For many languages, reliable language-specific literature search systems also exist, including SciELO (https://scielo.org/) for Spanish and Portuguese, J-STAGE (https://www.jstage.jst.go.jp/) for Japanese, KoreaScience (https://www.koreascience.or.kr/) for Korean, and CNKI (https://cnki.net/) for simplified Chinese. Using appropriate search strings in each language is key when searching for literature across multiple languages, although this would also require help from native speaker(s) of the language(s). Developing a list of relevant peer-reviewed non-English-language journals in each discipline, just as we did in this study (see S1 Data), could also help.

literature searching, established and adopted for the development of the Conservation Evidence database [17] and published elsewhere [38]. Discipline-wide literature searching involves first identifying literature sources (peer-reviewed academic journals in our case) that are likely to contain relevant information, and manually scanning titles and abstracts (or summaries) of every document in those sources. We adopted discipline-wide literature searching, rather than systematic mapping/reviewing, as the former approach does not depend on search term choice and can identify novel conservation interventions that would not necessarily have been identified on the basis of predetermined criteria for study inclusion [39]. For more details on the Conservation Evidence database, see section **English-language studies on the effectiveness of conservation interventions**. Although non-English-language grey literature (reports, theses, etc.) could also play an important role in environmental evidence syntheses [29], our searches focused only on studies published in peer-reviewed journals, so as to enable a comparison between eligible non-English-language studies and peer-reviewed English-language studies stored in the Conservation Evidence database.

**Selecting languages.**   We originally aimed to cover the top 15 non-English languages on the basis of the number of conservation-related publications, provided in S1 Table of [16]. However, we could not find native speakers of Swedish and Dutch who were willing to collaborate, and, thus, both languages were excluded from our searches. Instead, we were able to cover 3 additional languages (Arabic, Hungarian, and Ukrainian). In total, our searches covered 16 languages (S2 Table).

**Searchers.**   Our searches were conducted by a total of 38 native speakers of the 16 languages covered (hereafter referred to as searchers). The number of searchers for each language ranged from 1 to 6 (see S2 Table for more detail). We used a range of approaches (e.g., known networks, social media, e-mail lists, and the website of the translatE project: https://translatesciences.com/) to recruit our searchers. The searchers were required to be at least undertaking or have a bachelor's degree, but often had higher research (i.e., master's or doctorate) degrees, in a relevant discipline (ecology, biodiversity conservation, etc.), to ensure that they could fully understand the relevant studies and assess their eligibility during screening.

Before starting the searches, every searcher was trained through the following 4 steps. First, searchers were directed to read through a guidance document detailing the objectives and

processes of the searches. Second, searchers were also requested to read and understand the full criteria for selecting eligible studies during the searches, which were described in detail, together with examples of 14 eligible and 5 noneligible English-language studies, each with a full explanation on why it was or was not eligible. Third, searchers were advised to visit the Conservation Evidence website, particularly the page providing training resources (https:// www.conservationevidence.com/content/page/89), and familiarise themselves with eligible English-language studies that tested the effectiveness of conservation interventions (listed at: https://www.conservationevidence.com/data/studies). Finally, all searchers were asked to conduct a test study screening, where they were requested to read the metadata (publication year, journal, volume, issue, authors, title, and abstract) of 51 English-language papers (29 from volume 200 (2016) of Biological Conservation, and 22 from volume 30 (2016) of Conservation Biology), which included a total of 14 eligible studies, decide if each study was eligible or not, and provide the full reasoning for their decisions. The outcome of the test screening was examined by either TA, VB-E, or other members of the Conservation Evidence project, who provided searchers with feedback.

**Identifying and selecting journals for each language.** We first identified and listed peer-reviewed academic journals published in each language, which were likely to contain eligible studies. This process involved 1 to 4 researchers for each language (all native speakers of the target language, with at least a bachelor's, and often higher, research degree; this often included the searchers) from relevant disciplines (see S2 Table for more detail), who used a range of approaches (personal knowledge, opinions from colleagues, local literature databases, web searches, etc.) to identify as many potentially relevant journals as possible. All journals identified were then grouped into 3 categories: "very relevant" (often journals in ecology and biodiversity conservation, as well as taxonomic journals, such as those in ornithology, mammalogy, herpetology, plant sciences, etc.), "relevant" (mostly journals in relevant disciplines, such as agricultural/forest sciences and general zoology), and "maybe relevant" (all others). Subsequent searches aimed to at least cover all journals categorised as "very relevant" and, when possible, those in the other 2 categories (see S1 Data for the list of all journals searched).

**Screening papers in each journal.** Searches for eligible studies in each journal were conducted by manually scanning the title and abstract (or summary) of every peer-reviewed non-English-language paper published in the journal and by reading the main text of all papers for which the title and/or abstract were suggestive of fulfilling the eligibility criteria (fully described below). All papers that appeared to meet the eligibility criteria were identified as potentially eligible studies, with the relevant metadata recorded (see Data coding), and were then passed on to the validation process (see Study validity assessment). The journals were searched backwards from the latest volume, either to the earliest published volume or going back 10 years for long-running journals (see S1 Data for publication years covered for each journal). We also recorded the total number of papers screened in each journal.

The following eligibility criteria, which were developed and published by the Conservation Evidence project (https://osf.io/mz5rx/), were used.

## Criteria A: Include studies that measure the effect of an intervention that might be done to conserve biodiversity

1. Does this study measure the effect of an intervention that is or was under the control of humans, on wild taxa (including captives), habitats, or invasive/problem taxa? If yes, go to 3. If no, go to 2.

2. Does this study measure the effect of an intervention that is or was under the control of humans, on human behaviour that is relevant to conserving biodiversity? If yes, go to **Criteria B**. If no, the study will be excluded.

3. Could the intervention be put in place by a conservationist/decision maker to protect, manage, restore or reduce impacts of threats to wild taxa or habitats, or control or mitigate the impact of the invasive/problem taxon on wild taxa or habitats? If yes, the study will be included. If no, the study will be excluded.

- Eligible populations or subjects
  Included: Individuals, populations, species, or communities of wild taxa, habitats or invasive/problem taxa.
  Excluded: Domestic/agricultural species.

- Eligible interventions
  Included: Interventions that are carried out by people and could be put in place for conservation. Interventions within the scope of the searches include, but not limited to:

· Clear management interventions, e.g., closing a cave to tourism, prescribed burning, mowing, controlling invasive species, creating or restoring habitats;

· International or national policies;

· Reintroductions or management of wild species in captivity; and

· Interventions that reduce human–wildlife conflict.
  See **Criteria B** for interventions that have a measured outcome on human behaviour only. Also see https://www.conservationevidence.com/data/index for more examples of interventions.
  Excluded: Impacts of threats (interventions that remove threats would be included), impacts from natural processes (e.g., tree falls, natural fires), and impacts from background variation (e.g., soil type, vegetation, climate change).

- Eligible outcomes
  Included: Any outcome (can be negative, neutral, or positive, does not have to be statistically significant) that is quantified and has implications for the health of individuals, populations, species, communities or habitats, including, but not limited to:

· Individual health, condition or behaviour, including in captivity: e.g., growth, size, weight, stress, disease levels or immune function, movement, use of natural/artificial habitat/structure, range, predatory or nuisance behaviour that could lead to retaliatory action by humans.

· Breeding: egg/offspring/seed/sperm production, sperm motility/viability after freezing, natural/artificial breeding success, birth rate, offspring condition/survival, and overall recruitment.

· Genetics: genetic diversity, genetic suitability (e.g., adaptation to local conditions, use of flyways for migratory species).

· Life history: age/size at maturity, survival, mortality.

· Population measures: number, abundance, density, presence/absence, biomass, movement, cover, age-structure, species distributions (only in response to a human action), disease prevalence, and sex ratio.

· Community/habitat measures: species richness, diversity measures (including trait/functional diversity), community composition, community structure (e.g., trophic structure), area covered (e.g., by different habitat types), and physical habitat structure (e.g., rugosity, height, basal area).

- Eligible types of study design
Included: Studies with After, Before–After (BA), Control–Impact (CI), BACI, or RCT designs (using the definition provided in [40]). Literature reviews, systematic reviews, meta-analyses, or short notes that review studies that fulfil the eligibility criteria are also included. Studies that use statistical/mechanistic/mathematical models to analyse real-world data or compare models to real-world situations are also included (if they otherwise fulfil the eligibility criteria).
Excluded: Theoretical modelling studies, opinion pieces, correlations with habitat types where there is no test of a specific intervention by humans, or pure ecology (e.g., movement, distribution of species).

## Criteria B: Include studies that measure the effect of an intervention that might be done to change human behaviour for the benefit of biodiversity

1. Does this study measure the effect of an intervention that is or was under human control on human behaviour (actual or intentional), which is likely to protect, manage, restore, or reduce threats to wild taxa or habitats (including mitigating the impact of invasive/problem taxon on wild taxa or habitats)? If yes, go to 2. If no, the study will be excluded.

2. Could the intervention be put in place by a conservationist, manager, or decision maker to change human behaviour? If yes, the study will be included. If no, the study will be excluded.

- Eligible populations or subjects
Included: Actual or intentional human behaviour including self-reported behaviours. Change in human behaviour must be linked to outcomes for wild taxa or habitats.
Excluded: Human psychology (tolerance, knowledge, awareness, attitude, perceptions, or beliefs). Changes in behaviour linked to outcomes for human benefit, even if these occurred under a conservation program (e.g., we would exclude a study demonstrating increased school attendance in villages under a community-based conservation program).

- Eligible interventions
Included: Interventions that are under human control and change human behaviour, resulting in the conservation, management, and restoration of wild taxa or habitats. Interventions that are particularly likely to have a behaviour change outcome include, but are not limited to:

· Enforcement: hunting restrictions, market inspections, increase number of rangers, patrols or frequency of patrols in, around or within protected areas, improve fencing/physical barriers, improve signage.

· Behaviour change: promote alternative/sustainable livelihoods, payment for ecosystem services, ecotourism, poverty reduction, increased appreciation or knowledge, debunking misinformation, altering or reenforcing local taboos, financial incentives.

· Governance: protect or reward whistle-blowers, increase government transparency, ensure independence of judiciary, provide legal aid.

· Market regulation: trade bans, taxation, supply chain transparency laws.

· Consumer demand reduction: increase awareness or knowledge, fear appeals (negative association with undesirable product), benefit appeal (positive association with desirable behaviour), worldview framing, moral framing, employing decision defaults, providing decision support tools, simplifying advice to consumers, promoting desirable social norms, legislative prohibition.

· Sustainable alternatives: certification schemes, artificial alternatives, sustainable alternatives.

· New policies for conservation/protection.
  Excluded: Impacts from climatic or other natural events. Studies with no intervention, e.g., correlating human personality traits with likelihood of conservation-related behaviours.

• Eligible outcomes
  Included: Any human behaviour outcome (can be negative, neutral, or positive, does not have to be statistically significant) that is quantified and is likely to have an outcome on wild taxa or habitats, including, but not limited to:

· Change in adverse behaviours (which directly threaten biodiversity), e.g., unsustainable hunting, burning, grazing, urban encroachment, creating noise, entering sensitive areas, polluting or dumping waste, clearing or habitat destruction, introducing invasive species.

· Change in positive behaviours, e.g., uptake of alternative/sustainable livelihoods, number of households adopting sustainable practices, donations.

· Change in policy or conservation methods, e.g., placement of protected areas, protection of key habitats/species.

· Change in consumer or market behaviour, e.g., purchasing, consuming, buying, willingness to pay, selling, illegal trading, advertising, consumer fraud.

· Behavioural intentions to do any of the above.

• Eligible types of study design
  Same as **Criteria A**.

**Data coding.**   From each of the studies that were identified by searchers as potentially eligible, the following metadata were extracted and recorded using a template file:

• Journal language

• Journal publication country

• Reference type (either original article, review, short note or others)

• Authors

• Publication year

• Title in English (if available) and in the non-English language

• Journal name in English (if available) and in the non-English language

• Volume/Issue/Pages

- Abstract in English (if available) and in the non-English language

- Keywords in English (if available) and in the non-English language

- Link to the article (URL if available)

- Study site locations (coordinates; mean coordinates where a study had multiple sites, or city/state/province/country if coordinates were not available)

- Study design (either After, BA, CI, BACI, RCT, or review; using the definition of each design provided in [40])

- Broad species group(s)/habitat(s) studied

- Scientific name of study species (if available)

- Common name of study species in English and in the non-English language (if available)

- One-sentence summary in the form of: "This study tested the effect of [*intervention(s)*] on [*measured outcome*] of [*target species or ecosystem(s)*]" (e.g., "This study tested the effect of providing nest boxes on the breeding success of blue tits")

The metadata were extracted largely by the searchers, but, for some languages where the searchers were not available, by other collaborators who are native speakers of the language and are at least undertaking or have a bachelor's, but often higher research, degree in a relevant discipline (see S2 Table for more details). They were all requested to first read and fully understand our guidance detailing the definitions of different study designs (provided in [40]) before starting data coding.

For all studies that were validated as eligible (see Study validity assessment), the recorded names of birds, mammals, and amphibians were standardised based on the lists of bird species names used by BirdLife International [41], and mammal and amphibian species names used by IUCN [42]. We focused on these 3 taxa for comparing study locations and species between languages because English-language studies testing the effectiveness of conservation interventions for these 3 taxa have extensively been searched using both discipline-wide literature searches and subject-wide evidence syntheses [17]. To identify species name synonyms we used the package "taxize" [43] in R [44] with API keys generated at the NCBI (https://www.ncbi.nlm.nih.gov/account/) and IUCN (https://apiv3.iucnredlist.org/api/v3/token) websites.

**Study validity assessment.** The eligibility of each study that was identified as being potentially eligible was validated by at least one experienced literature searcher (assessors) at the Conservation Evidence project (see S2 Table for more details), who regularly screen, identify, and summarise eligible studies using the same eligibility criteria (see Screening papers in each journal) but who mostly are not native speakers of each non-English language. This process was conducted by assessing the English-language title, abstract, and one-sentence summary of each study identified by the searchers (see Data coding), and, where the validity could not be determined easily, also involved direct discussions between the relevant searchers and assessors to obtain clarification on the details of each study. Those studies that were deemed ineligible by the assessors were excluded from the final list of eligible studies in each language.

**Limitations.** Although, as described above, we adopted a search strategy that allowed us to identify eligible studies in as unbiased a way as possible, our search results can still suffer from some inevitable limitations:

· Language selection

  Of the top 15 non-English languages on the basis of the number of conservation-related publications, our searches could not cover Swedish and Dutch. Nevertheless, we expect that

the exclusion of these 2 languages only had a minimal effect on our conclusions, as conservation-related publications in these 2 languages were estimated to only constitute 0.87% of publications in the top 15 non-English languages [16], while we also covered 3 additional languages.

· Journal selection

Although we identified 465 journals in 16 languages, we were only able to screen 326 of them, as we prioritised journals ranked as "very relevant" and "relevant" for some languages when there was a shortage of searchers and/or their time that could be dedicated to the search process. Therefore, we assessed whether our choice of journals screened in each language was appropriate for identifying the most eligible studies in the language, by examining the "rank-abundance" curve for each language, where the x-axis of the curve was the rank of searched journals according to the % of eligible studies (the journal with the highest % of eligible studies was given rank 1), and the y-axis was the % of eligible studies. If a curve reached zero (i.e., there were almost no eligible studies) in lower-ranked journals, we interpreted it as an indication that sufficient coverage of journals had been reached for that language (see S1 Fig for the result).

· Publication year selection

Searches for some long-running journals only went back 10 years from the latest volume, thus potentially missing some eligible studies dating back further. We thus assessed the effects of excluding earlier volumes of long-running journals from our searches, by testing how the number of eligible studies changed over time in each journal (see Fig 1 for the result).

· Possibility of missing eligible studies

We tried to identify as many eligible studies as possible in each language, by making sure that (i) every searcher was well qualified and trained before starting the searches (see Searchers); and (ii) when in doubt, searchers keep, rather than reject, a study as potentially eligible, the validity of which was later assessed by independent experts (see Study validity assessment). Nevertheless, we cannot dismiss the possibility that some eligible studies were missed during the searches. This would have caused a potential underestimation of the number of eligible studies published in non-English languages. However, this should not undermine our main conclusion that scientific evidence published in non-English languages could fill gaps in the geographic and taxonomic coverage of English-language evidence for conservation.

· Potential variations in assessment outcomes of eligible studies and study designs among searchers

Although we did our best to train searchers to fully understand the eligibility criteria (see Screening papers in each journal) and the definition of different study designs (see Data coding), some inevitable variations may remain in the assessment outcomes of eligible studies and study designs among searchers. This would potentially affect the reported patterns in (i) the number and proportion of eligible studies among non-English languages (S1 Fig); and (ii) the proportion of different study designs among different languages (Fig 2). Nevertheless, among-searcher variations in judgements should affect neither (i) yearly increases in the number of eligible non-English-language studies in each journal (Fig 1), as the same journal was searched by a single searcher, nor (ii) the spatial and taxonomic complementarity between English- and non-English-language studies (Figs 3 and 4), assuming that any such variations in assessment outcomes only affected a limited number of non-English-

language studies and thus have not drastically changed the overall patterns in the differences between English and non-English-language studies.

· Effects of publication bias

We focused only on studies published in peer-reviewed academic journals and thus did not consider the effects of publication bias, caused by ignoring grey literature, within each language, while recognising that important scientific knowledge may also be published in non-English-language grey literature [29]. Therefore, it should be noted that the conclusions of this study are limited to peer-reviewed studies published in academic journals.

## English-language studies on the effectiveness of conservation interventions

To compare study characteristics (i.e., study design, study location, and study species) between eligible English- and non-English-language studies, we used English-language studies stored in the Conservation Evidence database [17]. Those English-language studies were identified through a screening of peer-reviewed papers published in over 330 English-language academic journals including local and taxonomic journals (see the list at: https://www.conservationevidence.com/journalsearcher/english) based on the same eligibility criteria as described in the section "Screening papers in each journal" above (also see [17] for more details). We extracted the metadata (including publication year, study site coordinates—mean coordinates where a study had multiple sites, study design, scientific and common names of study species) for each of the 4,412 English-language studies (S4 Data) (including 284 studies on amphibians, 1,115 studies on birds, and 1,154 studies on mammals; S5 Data) from the database on December 11, 2020. Again, here, we defined a paper published in a peer-reviewed journal as a study. The Conservation Evidence database also stores some non-English-language peer-reviewed studies, most of which were identified incidentally by the project. Those non-English-language studies were also incorporated into our dataset of non-English-language studies, if they were in any of the 16 languages covered in this study (a total of 74 non-English-language studies; see records with "Source" being "Ad hoc" in S3 Data). For birds, mammals, and amphibians, species names were standardised using the lists of bird, mammal, and amphibian species names used by BirdLife International [41] and the IUCN [42].

## Analyses of eligible studies

**Comparing the proportion of eligible studies in each journal.**   We first calculated the proportion of eligible studies for each non-English-language journal, by dividing the number of eligible studies by the total number of studies screened in the journal. Journals with 30 or fewer studies screened were excluded from this calculation, as the estimated proportions would be unreliable given the small sample size.

**Testing yearly changes in the number of eligible non-English-language studies.**   To test whether the number of eligible non-English-language studies had changed over time, we focused only on journals with 10 or more eligible studies, resulting in journals/studies in a total of 12 languages (shown in Fig 1) being used in the following analysis. For each language, we fitted a generalised linear model (GLM) assuming a Poisson distribution with the number of eligible studies in each year in each journal as the response variable, and year and journal (for languages with more than one journal) as the explanatory variables. Journals were included in each model as a fixed, not random, effect, as the number of journals with 10 or more eligible studies in each language was relatively small (9 for Japanese, 5 for German, and <5 for all others), making it difficult to estimate the among-journal variance accurately in a mixed model [45].

**Comparing study designs.** To test whether there was a difference in study designs adopted between studies in different languages, we only included studies based on one of the following 5 designs: After, BA, CI, BACI, and RCT. These study designs were recorded as an ordinal variable with RCT being the least biased design, followed by BACI, CI, BA, and After, based on results from [24]. Considering that English-language studies in English-speaking countries (especially the United Kingdom and the United States) may adopt more robust study designs than English-language studies in other countries, English-language studies were further divided into 2 groups; studies conducted in countries where English is an official language ("English–official"), and studies in all other countries ("English–others"), using information on countries' official languages in [46]. We then fitted a cumulative link model using the ordinal package [47] in R, with ordered study designs in each study as the response variable and languages (16 non-English languages and "English–official," compared to "English–others" as the reference category) and taxa (birds, mammals, and others, compared to amphibians as the reference category) as the explanatory variables.

**Comparing study locations.** To test whether there was a systematic bias in study locations between English- and non-English-language studies, we first calculated the number of studies for each language in each $2° \times 2°$ grid cell. Studies without study coordinates were excluded from this calculation, leading to 4,254 English-language studies (including 267 studies on amphibians, 1,084 studies on birds, and 1,062 studies on mammals) and 1,202 non-English-language studies (including 53 studies on amphibians, 244 studies on birds, and 153 studies on mammals) being used in the following analysis. As the latest English-language studies on birds, amphibians, and mammals stored in the Conservation Evidence database were those published in 2011, 2012, and 2018, respectively, non-English-language studies after those years were excluded from the comparison of studies on each taxon, leading to 31 studies on amphibians, 182 studies on birds, and 146 studies on mammals being used in the analysis. We used a conditional autoregressive (CAR) model assuming a Poisson distribution to test the association between the number of non-English-language studies (the response variable) and the number of English-language studies (the explanatory variable) within each grid cell while accounting for spatial autocorrelation in residuals (see **Data availability** for the availability of the R code). We fitted the model to the data with the Markov chain Monte Carlo (MCMC) method in OpenBUGS 3.2.3 [48] and the R2OpenBUGS package [49] in R. We set prior distributions of parameters as noninformatively as possible, so as to produce estimates similar to those generated by a maximum likelihood method; we used an improper uniform distribution (i.e., a uniform distribution on an infinite interval) for the intercept following [50], a normal distribution with a mean of 0 and variance of 100 for the coefficient of the explanatory variable, and Gamma distributions with a mean of 1 and variance of 100 for the inverse of variance in an intrinsic Gaussian CAR distribution. We ran each MCMC algorithm with 3 chains with different initial values for 35,000 iterations with the first 5,000 discarded as burn-in and the reminder thinned to 1 in every 12 iterations to save storage space. Model convergence was checked with R-hat values.

**Comparing species.** To test whether there was a systematic bias in study species between English- and non-English-language studies, we first calculated the number of English- and non-English-language studies available for each species. We used generalised linear mixed models (GLMMs) assuming a Poisson distribution to test the association between the number of non-English-language studies (the response variable) and the number of English-language studies (the explanatory variable) for each species while accounting for phylogenetic autocorrelation by incorporating the family of each species as a random factor (see **Data availability** for the availability of the R code). The GLMMs were implemented in R with the lme4 package [51].

Other R packages used in the analyses and data visualisation were the following: data.table [52], dplyr [53], gridExtra [54], mapdata [55], mcmcplots [56], MCMCvis [57], plyr [58], RColorBrewer [59], rgdal [60], readxl [61], tidyverse [62], viridis [63], and writexl [64].

## Supporting information

**S1 Table. Results of a cumulative link model aimed at testing the association between ordered study designs (with RCT as the least biased design, followed by BACI, CI, BA, and After) in each study as the response variable, and languages (16 non-English languages and "English–official" (English-language studies conducted in countries where English is an official language), compared to "English–others" (English-language studies conducted in the other countries) as the reference category) and taxa (birds, mammals, and others, compared to amphibians as the reference category) as the explanatory variables.** Significant results are shown in bold. BA, Before–After; BACI, Before–After–Control–Impact; CI, Control–Impact; RCT, Randomised Controlled Trial.
(DOCX)

**S2 Table. List of those involved in searches and their roles for each language covered in this study.**
(DOCX)

**S1 Fig. The proportion (%) of eligible studies testing the effectiveness of conservation interventions in each journal in 16 non-English languages.** Coloured dots connected with a line represent all journals screened for each language, in decreasing order of % eligible studies; the journal with the highest % eligible studies is shown on the far left, while the journal with the lowest % eligible studies is on the far right. This figure was created using S1 Data and Code 5.
(DOCX)

**S2 Fig. The proportion of non-English-language studies (all 16 languages combined) to all studies (i.e., non-English and English-language studies combined) testing the effectiveness of conservation interventions within each 2˚ × 2˚ grid cell.** This figure was created using S3 and S4 Data with Code 3. Map produced from the Natural Earth dataset (v.4.1.0) at 1:50 m scale (https://www.naturalearthdata.com/downloads/50m-cultural-vectors/).
(DOCX)

**S3 Fig. The location of 31 non-English-language studies testing the effectiveness of conservation interventions for amphibian species (published in 2012 or earlier), compared to the number of English-language studies on amphibians within each 2˚ × 2˚ grid cell (133 grid cells in total).** Non-English-language studies were found in 23 grid cells, 16 of which were without any English-language studies (grid cells in black). The inset is a hexbin chart showing a significantly negative relationship between the number of English-language studies (No. English studies) and the number of non-English-language studies (No. non-English studies) within each grid cell. Brighter colours indicate more grid cells in each hexagon. This figure was created using S3 and S4 Data with Code 6. Map produced from the Natural Earth dataset (v.4.1.0) at 1:50 m scale (https://www.naturalearthdata.com/downloads/50m-cultural-vectors/).
(DOCX)

**S4 Fig. The location of 182 non-English-language studies testing the effectiveness of conservation interventions for bird species (published in 2011 or earlier), compared to the number of English-language studies on birds within each 2˚ × 2˚ grid cell (373 grid cells in**

**total).** Non-English-language studies were found in 75 grid cells, 59 of which were without any English-language studies (grid cells in black). The inset is a hexbin chart showing a significantly negative relationship between the number of English-language studies (No. English studies) and the number of non-English-language studies (No. non-English studies) within each grid cell. Brighter colours indicate more grid cells in each hexagon. This figure was created using S3 and S4 Data with Code 6. Map produced from the Natural Earth dataset (v.4.1.0) at 1:5 0m scale (https://www.naturalearthdata.com/downloads/50m-cultural-vectors/). (DOCX)

**S5 Fig. The location of 146 non-English-language studies testing the effectiveness of conservation interventions for mammal species (published in 2018 or earlier), compared to the number of English-language studies on mammals within each 2˚ × 2˚ grid cell (514 grid cells in total).** Non-English-language studies were found in 89 grid cells, 61 of which were without any English language studies (grid cells in black). The inset is a hexbin chart showing a significantly negative relationship between the number of English-language studies (No. English studies) and the number of non-English-language studies (No. non-English studies) within each grid cell. Brighter colours indicate more grid cells in each hexagon. This figure was created using S3 and S4 Data with Code 6. Map produced from the Natural Earth dataset (v.4.1.0) at 1:50 m scale (https://www.naturalearthdata.com/downloads/50m-cultural-vectors/). (DOCX)

**S6 Fig. The distribution of the number of English-language studies for each threatened amphibian, bird, and mammal species (blue), with species ranked on the x-axis in order of decreasing number of English-language studies, and the number of non-English-language studies per species for those threatened species studied by both English- and non-English-language studies (orange), and those studied only by non-English-language studies (red).** Note that a threatened mammal species with 38 English-language studies is not shown as an outlier. The insets are hexbin charts showing the relationship between the number of English-language studies (No. English studies) and the number of non-English-language studies (No. non-English studies) for each threatened species. Species classified as threatened (Critically Endangered, Endangered, or Vulnerable) based on IUCN. Brighter colours indicate more species in each hexagon. Only studies published in 2012 or earlier for amphibians, 2011 or earlier for birds, and 2018 or earlier for mammals were used in this figure. This figure was created using S3 and S5 Data with Code 4 and IUCN species lists available at: https://www.iucnredlist.org/resources/spatial-data-download. (DOCX)

**S1 Data. The list of non-English-language peer-reviewed journals related to biodiversity conservation identified in this study.** The explanations of column names are as follows: Language: journal publication language, Country: journal publication country, Journal title in English: journal title in English, Journal title in non-English language: journal title in the non-English language, First publication year: the first publication year, Latest publication year: the latest publication year (as of March 2021), Link (URL): link to the journal website, Research areas/taxa: broad research area and taxa covered in the journal, Searcher: searcher name, Years screened first: the publication year of the first volume screened, Years screened last: the publication year of the last volume screened, Years screened total: the number of years screened, Volumes screened: the number of volumes screened, Number of papers screened: the number of studies screened, Number of papers id as relevant by collaborators: the number of studies initially identified as eligible by searchers, Number of papers validated as relevant: the number

of studies validated as eligible, Number of papers added ad hoc from CE dataset: the number of studies added from the Conservation Evidence database, Total relevant: the total number of eligible studies, Comments: any other relevant notes.
(CSV)

**S2 Data. The list of 1,234 non-English-language studies identified as eligible in this study.** The explanations of column names are as follows: Paper ID: study ID, Translator Name: searcher name, Language: study publication language, Journal Country: journal publication country, Reference Type: the type of publications (journal article, review, etc.), Authors (separate with//): the name of authors, Year: publication year, Title–English: title in English, Title–non-English language: title in the non-English language, Journal: journal name, Volume: volume, Issue: issue, Pages: pages, Abstract–English: abstract in English, Abstract–non-English: abstract in the non-English language, Keywords–English: keywords in English, Keywords–non-English: keywords in the non-English language, Broad species group(s)/ habitat(s)/ ecosystem service(s): broad species group(s) / habitat(s) studied, Species Scientific Name: scientific name of study species, Species English Name: common name of study species in English, Species Non-English Name: common name of study species in the non-English language, Study design: study design adopted, Mean Lat: mean latitude of the study site(s), Mean Long: mean longitude of the study site(s), City/state or province/country: city/state/province/country of the study site(s), DOI: Digital Object Identifier, Link (URL): link to the paper.
(CSV)

**S3 Data. The list of species studied in the 1,234 non-English-language eligible studies.** The explanations of column names are as follows: Paper ID: study ID, Language: study publication language, IUCN: scientific name of study species used by the International Union for Conservation of Nature (IUCN name), Species Scientific Name: scientific name of study species recorded by searchers, Common name: common name of study species identified with the package 'taxize' in R, Species English Name: common name of study species in English recorded by searchers, Species Non-English Name: common name of study species in the non-English language recorded by searchers, Taxa: taxonomic group identified based on the IUCN name, Broad species group(s)/ habitat(s)/ ecosystem service(s): broad species group(s) / habitat(s) studied, Study design: study design adopted, Mean Lat: mean latitude of the study site(s), Mean Long: mean longitude of the study site(s), City/state or province/country: city/state/province/country of the study site(s), Journal: journal name, Journal Country: journal publication country, Source: the method of identifying the study (systematic review: discipline-wide literature searching, Ad hoc: identified in the Conservation Evidence database) Year: publication year.
(CSV)

**S4 Data. The list of species studied in the 4,412 English-language studies stored in the Conservation Evidence database.** The explanations of column names are as follows: rowed: record ID, pageid: study ID, journal_match_scimago: journal name used in the Scimago Journal Rank, journal: journal name, syn: Conservation Evidence synopsis including the study, int: conservation intervention tested, before: if the study has a Before element, controlled: if the study has control(s), randomised: if replications are randomised, review: if the study is based on a review or not, pubdate: publication year, lat: latitude of the study site(s), long: longitude of the study site(s), country: country of the study site(s), species: specific name of the study species, genus: generic name of the study species, family: family of the study species, order: order of the study species, class: class of the study species, binom: scientific name of the study species, pubtype: study publication type, original_title: paper title, ref_startpage: start page of the

paper, ref_endpage: end page of the paper, ref_vol: volume published, ref_issue: issue published, ref_doi: Digital Object Identifier, ref_citation: paper citation, ref_authorstring: authors. (CSV)

**S5 Data. The list of amphibian, bird, and mammal species studied in the English-language studies stored in the Conservation Evidence database.** The explanations of column names are as follows: pageid: study ID, syn: Conservation Evidence synopsis including the study, int: conservation intervention tested, before: if the study has a Before element, controlled: if the study has control(s), randomised: if replications are randomised, review: if the study is based on a review or not, pubdate: publication year, lat: latitude of the study site(s), long: longitude of the study site(s), country: country of the study site(s), species: specific name of the study species, genus: generic name of the study species, family: family of the study species, order: order of the study species, class: class of the study species, binom: scientific name of the study species (standardised based on the names used by the International Union for Conservation of Nature), habitat: broad habitat type studied, authors: authors, journal: journal name. (CSV)

**S1 Alternative Language Abstract. Alternative Language Abstract in French.** (PDF)

**S2 Alternative Language Abstract. Alternative Language Abstract in German.** (PDF)

**S3 Alternative Language Abstract. Alternative Language Abstract in Hungarian.** (PDF)

**S4 Alternative Language Abstract. Alternative Language Abstract in Italian.** (PDF)

**S5 Alternative Language Abstract. Alternative Language Abstract in Japanese.** (PDF)

**S6 Alternative Language Abstract. Alternative Language Abstract in Korean.** (PDF)

**S7 Alternative Language Abstract. Alternative Language Abstract in Persian.** (PDF)

**S8 Alternative Language Abstract. Alternative Language Abstract in Polish.** (PDF)

**S9 Alternative Language Abstract. Alternative Language Abstract in Portuguese.** (PDF)

**S10 Alternative Language Abstract. Alternative Language Abstract in Romanian.** (PDF)

**S11 Alternative Language Abstract. Alternative Language Abstract in Russian.** (PDF)

**S12 Alternative Language Abstract. Alternative Language Abstract in simplified Chinese.** (PDF)

**S13 Alternative Language Abstract. Alternative Language Abstract in Spanish.** (PDF)

**S14 Alternative Language Abstract. Alternative Language Abstract in traditional Chinese.**
(PDF)

**S15 Alternative Language Abstract. Alternative Language Abstract in Turkish.**
(PDF)

**S16 Alternative Language Abstract. Alternative Language Abstract in Ukrainian.**
(PDF)

## Acknowledgments

Thanks to I. Mangold and H. Korn for their help during the data collection, and M. Amano for all the support.

This communication reflects only the authors' view, and any of the funders is not responsible for any use that may be made of the information it contains.

## Author Contributions

**Conceptualization:** Tatsuya Amano.

**Formal analysis:** Tatsuya Amano, Violeta Berdejo-Espinola, Alec P. Christie.

**Funding acquisition:** Tatsuya Amano, Alec P. Christie, András Báldi, Sandro Bertolino, Kerstin Jantke, Joanna Kajzer-Bonk, Igor Khorozyan, Matthias-Claudio Loretto, Valentina Marconi, Juan P. Narváez-Gómez, Pablo Jose Negret, A. Nayelli Rivera-Villanueva, Marie-Morgane Rouyer, Flóra Vajna, José Osvaldo Valdebenito, Rafael D. Zenni, William J. Sutherland.

**Investigation:** Tatsuya Amano, Violeta Berdejo-Espinola, Alec P. Christie, Munemitsu Akasaka, András Báldi, Sandro Bertolino, Min Chen, Chang-Yong Choi, Magda Bou Dagher Kharrat, Luis G. de Oliveira, Perla Farhat, Marina Golivets, Nataly Hidalgo Aranzamendi, Kerstin Jantke, Joanna Kajzer-Bonk, M. Çisel Kemahlı Aytekin, Igor Khorozyan, Kensuke Kito, Ko Konno, Da-Li Lin, Yang Liu, Yifan Liu, Matthias-Claudio Loretto, Valentina Marconi, Juan P. Narváez-Gómez, Pablo Jose Negret, Elham Nourani, Jose M. Ochoa Quintero, Rachel Rui Ying Oh, Ana C. Piovezan-Borges, Ingrid L. Pollet, Danielle L. Ramos, Ana L. Reboredo Segovia, A. Nayelli Rivera-Villanueva, Marie-Morgane Rouyer, Richard Schuster, Dominik Schwab, Çağan H. Şekercioğlu, Hae-Min Seo, Yushin Shinoda, Shan-dar Tao, Ming-shan Tsai, Flóra Vajna, José Osvaldo Valdebenito, Svetlana Vozykova, Paweł Waryszak, Veronica Zamora-Gutierrez, Rafael D. Zenni, Wenjun Zhou.

**Methodology:** Tatsuya Amano, Violeta Berdejo-Espinola, Alec P. Christie, William J. Sutherland.

**Project administration:** Tatsuya Amano, Violeta Berdejo-Espinola, Kate Willott, Elizabeth H. M. Tyler, William J. Sutherland.

**Supervision:** Tatsuya Amano, William J. Sutherland.

**Validation:** Tatsuya Amano, Violeta Berdejo-Espinola, Kate Willott, Anna Berthinussen, Andrew J. Bladon, Nick Littlewood, Philip A. Martin, William H. Morgan, Nancy Ockendon, Silviu O. Petrovan, Ricardo Rocha, Katherine A. Sainsbury, Gorm Shackelford, Rebecca K. Smith.

**Visualization:** Tatsuya Amano, Alec P. Christie.

**Writing – original draft:** Tatsuya Amano.

**Writing – review & editing:** Tatsuya Amano, Violeta Berdejo-Espinola, Alec P. Christie, Kate Willott, Munemitsu Akasaka, András Báldi, Anna Berthinussen, Sandro Bertolino, Andrew J. Bladon, Min Chen, Chang-Yong Choi, Magda Bou Dagher Kharrat, Luis G. de Oliveira, Perla Farhat, Marina Golivets, Nataly Hidalgo Aranzamendi, Kerstin Jantke, Joanna Kajzer-Bonk, M. Çisel Kemahlı Aytekin, Igor Khorozyan, Kensuke Kito, Ko Konno, Da-Li Lin, Nick Littlewood, Yang Liu, Yifan Liu, Matthias-Claudio Loretto, Valentina Marconi, Philip A. Martin, William H. Morgan, Juan P. Narváez-Gómez, Pablo Jose Negret, Elham Nourani, Jose M. Ochoa Quintero, Nancy Ockendon, Rachel Rui Ying Oh, Silviu O. Petrovan, Ana C. Piovezan-Borges, Ingrid L. Pollet, Danielle L. Ramos, Ana L. Reboredo Segovia, A. Nayelli Rivera-Villanueva, Ricardo Rocha, Marie-Morgane Rouyer, Katherine A. Sainsbury, Richard Schuster, Dominik Schwab, Çağan H. Şekercioğlu, Hae-Min Seo, Gorm Shackelford, Yushin Shinoda, Rebecca K. Smith, Shan-dar Tao, Ming-shan Tsai, Elizabeth H. M. Tyler, Flóra Vajna, José Osvaldo Valdebenito, Svetlana Vozykova, Paweł Waryszak, Veronica Zamora-Gutierrez, Rafael D. Zenni, Wenjun Zhou, William J. Sutherland.

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
