## [Editor Report · Decision Letter 0]

7 Jun 2021

Dear Dr Amano, 

Thank you for submitting your manuscript entitled "Tapping into non-English-language science for the conservation of global biodiversity" for consideration as a Research Article by PLOS Biology.

Your manuscript has now been evaluated by the PLOS Biology editorial staff, as well as by an academic editor with relevant expertise, and I'm writing to let you know that we would like to send your submission out for external peer review.

IMPORTANT: Your paper would be better considered as a Meta-Research Article. No re-formatting is needed, but please could you select "Meta-Research Article" as the article type when you upload your additional meta-data (see next paragraph)?

Please re-submit your manuscript within two working days, i.e. by Jun 09 2021 11:59PM.

Kind regards,

Roli Roberts

Roland Roberts

Senior Editor

PLOS Biology

rroberts@plos.org

---

## [Decision Letter · Decision Letter 1]

27 Jul 2021

Dear Dr Amano,

Thank you for submitting your manuscript "Tapping into non-English-language science for the conservation of global biodiversity" for consideration as a Meta-Research Article by PLOS Biology. As with all papers reviewed by the journal, yours was evaluated by the PLOS Biology editors as well as by an Academic Editor with relevant expertise and by independent reviewers. I have taken over the handling of your submission in the absence of my colleague, Roli Roberts, in order to prevent any unnecessary loss of time. As you will see from the reports at the end of this email, the reviewers are highly supportive of the work and commended the strength, originality and quality of the data and analysis. 

Based on the reviews, we are likely to accept this manuscript for publication, provided you satisfactorily address all points raised by the reviewers, which mainly request textual changes to clarify, provide further context or caveat the findings. In this regard, both reviewers #1 and #2 ask that you propose solutions for the problems that you identify, which we think should be taken on board. We would suggest a good way might be to add a Box in the Discussion in which you highlight the practical challenges of accessing non-english language articles, and suggest some ways forward. Finally, our Academic Editor thought it worth emphasising in the manuscript that the extent to which the conclusion of the current survey applies to other areas in biology might vary hugely among fields. By its very nature, a lot of conservation work is local and the focus is therefore on communicating with regional stakeholder rather than the international community, which might favour using the local language. In contrast, studies on, say, theory driven ecology or evolutionary biology are inherently more international in flavour. This point should perhaps also be mentioned in the Box.

Please also make sure to address the following editorial policy-related request:

- DATA POLICY:

You may be aware of the PLOS Data Policy, which requires that all data be made available without restriction: http://journals.plos.org/plosbiology/s/data-availability. 

We note that you have deposited all raw data and code at OSF, but in addition we ask that all individual quantitative observations that underlie the data summarized in the main and supplementary figures be made available, clearly identifiable as such, and that each figure legend calls out to the location where the data specifically underlying the figure can be found. This can be done in one of the following forms:

Regardless of the method selected, please ensure that you provide the individual numerical values that underlie the summary data displayed in all the main and supplementary figures, as they are essential for readers to assess your analysis and to reproduce it.

----

As you address the above items, please take this last chance to review your reference list to ensure that it is complete and correct. If you have cited papers that have been retracted, please include the rationale for doing so in the manuscript text, or remove these references and replace them with relevant current references. Any changes to the reference list should be mentioned in the cover letter that accompanies your revised manuscript.

We consider that, as the changes required are minor, we can reasonably expect the manuscript back in two weeks. We do note however that things may slow down over the summer, so please let us know if you would need more time to submit your revision. 

*Published Peer Review History*

*Early Version*

With best wishes,

Nonia

Nonia Pariente, PhD

Editor in Chief

PLOS Biology

npariente@plos.org

for

Senior Editor,

rroberts@plos.org,

PLOS Biology

----

Reviewer remarks:

Reviewer #1: This is a fascinating study in which the authors quantify the contribution of non-English-language studies to evidence synthesis and evaluate the support for four common (mis)perceptions regarding studies written in languages other than English. Those perceptions are: (1) that the additional evidence found in non-English-language studies is negligible; (2) that the number of non-English-language studies is decreasing; (3) that the rigorousness of non-English-language studies is lower; (4) that the evidence in non-English-language studies is a subset of the evidence in English-language studies. 

Based on their results, the authors show that the data somewhat support only the third perception and that non-English-language studies contain essential information (that is not always found in English-language studies), and which can help reduce the current geographic and taxonomic biases in conservation. Overall, the study is very well written, and the methods and analysis robust. It addresses an important topic, and it could make a useful contribution to the literature. 

My only major comment—and I understand that this may go beyond the scope of this particular study—is that the authors do not provide any specific solutions/recommendations regarding how this problem could be solved. 

For example, although the authors give some very specific recommendations about how to support local scientists to improve rigorousness (e.g., through the distribution of free teaching materials; lines 323-325), they provide no recommendations about how other researchers can incorporate non-English-language studies in their analyses.

In fact, the manuscript is written and presented as if the main (only?) reason for the lack of inclusion of non-English-language studies is because they are considered non-essential. For example, the abstract begins by arguing that many scientists assume that the evidence found in English-language studies is sufficient, and the analysis around the four perceptions is designed to show the opposite. There is no mention of the significant difficulties associated with including non-English-language studies (i.e., even in cases in which researchers are well aware of their importance). 

For example, I would argue that another important reason why non-English-language studies are often excluded is that many times researchers don't have access to them, e.g., when they do not speak the particular language. The authors themselves were unable to assess two of the top-15 languages for that same reason. And to assess the rest of the 16 languages, a large team of "searchers" was needed, something that may not always be feasible. 

Therefore, with the above in mind, I was somewhat surprised to see that such limitations were not acknowledged (and discussed) in the manuscript. Considering the difficulties associated with incorporating non-English-language studies, I believe the manuscript could be more useful to the scientific community if the authors were able to provide some practical recommendations regarding how to overcome such limitations…based on the authors' knowledge and experience on the topic. Such recommendations could range from simple advice on locating foreign journals and approaching local scientists to more complex suggestions, e.g., regarding how to prioritize which languages are likely to be more important. 

Other minor comments:

Lines 146-147: the percentages given here are difficult to interpret without knowing the actual methods. For example, what does it mean to "expand taxonomic coverage" by 32%? Perhaps this section could be rephrased in a way that makes the percentages clearer? 

Lines 199-200: Similarly, the description of the third perception here is somewhat vague. There are different ways to assess the quality of a study, and it is not clear how it was done until one reads the corresponding section in the methods (which is towards the very end of the manuscript). I am wondering if a few more details could be given here since the quality was assessed in a particular way, and the results could differ if a different measure was selected.

Lines 212-218: These numbers are very interesting and show that there are apparent differences between the countries. Could these results be used to provide some basic suggestions for best practices for researchers working at the global level, e.g., in the sense that even if researchers can't include all non-English-language studies, it is essential that they at least evaluate studies in Japanese, for instance? I understand that this may not be possible since these numbers are likely to vary depending on the context and the purpose of each study. At the same time, if there was an obvious way for researchers to prioritize which languages to assess, it could make the task less overwhelming and more probable.

Line 235-237. I am probably missing something here, but I do not see the connection between sample size (i.e., number of studies per taxonomic group) and study design.

Reviewer #2: It is not every day that I get to review such important work. This study investigated four assumptions that have never been investigated at this scale: 1) non-English studies are negligible, 2) the number of relevant non-English studies are decreasing, 3) non-English studies are lesser quality than English ones and 4) English studies are representative of all studies. To my surprise (or probably to almost all), only one of them was supported; at least, it seems to be the case in the literature on the effectiveness of conservation intervention. This study is based on a huge effort, robust data and analysis. Further, the text is concisely and informatively written with wonderful visualization. It is very unusual for me not to suggest, but I cannot really think of anything to add or mend. 

However, if the editor agrees to do this, I have one request. This article has convinced me of the importance of non-English literature, and that is the main conclusion they draw. However, this is a problem for many people who do literature synthesis work, which means most scientists these days (we all need to synthesis the relevant literature). This work was based on an extensive international network of collaborators that enabled the authors to conduct this impressive work. But many of us do not have such a network. I would like to see some discussion on how to get such an international network for many of us to conduct a synthesis work that incorporates non-English literature (actually, a box will be a perfect one for this job). I am after some tips and difficulties to do such collaborations. 

Reviewer #3: This is a novel, interesting and timely article that highlights the wealth of informaiton available about conservation interventions and their effectiveness that can be overlooked in non-english articles. It builts upon previous work by the same lead author to show that not only are many biodiversity articles published in non-english sources, but that including these can broaden the taxonomic and geographical coverage of meta-analyses.

The manuscript is already clear and very well written and I only have a few minor queries/suggestions:

Line 197: 'rarely corroborated together' - this phrase

Line 220-221: 'six out of the 12 languages' - If the authors wish to stress this, it would be nice to have these listed in parentheses here (as is done for robustness further down) rather than just highlighting Portugese and Russian.

Line 239: If the manuscript is to be presented with the methods further down, it might be nice to have a small sentence here introducing the 2x2 deg. grid cells, or just refer to the methods perhaps?

Line 239: Also - I wonder if using a 2 x 2 deg grid cell underestimates the increase in geographical coverage of the new articles. The representative area covered by a grid cell in temperate areas is smaller than at the tropics and many of the new grids cells in Fig3 are tropical. I realise this isnt the thrust of this bit of the analysis, but the increase in coverage might be larger that the percentage of cells suggests.

Line 253: I was left wondering if any of the studies touched on DD species, which seems unlikely, but it might be nice to point out if true. If not, highlighting that non-english searches can reveal any informaiton on DD species would be very useful.

Line 366: I think it would be important to point out this limitation in the main text - It is understandable that this study focusses on peer-reviewed literature, but I feel that higlighting that the potential role of evidence from grey literature remains untested is important.

Line 984: reads a little oddly because 'English - others' bleeds into ' - others are shown with pink', perhaps italicise the names of the groups (or quote them?)

---

## [Editor Report · Decision Letter 2]

25 Aug 2021

Dear Dr Amano,

On behalf of my colleagues and the Academic Editor, Michael Jennions, I'm pleased to say that we can in principle offer to publish your Meta-Research Article "Tapping into non-English-language science for the conservation of global biodiversity" in PLOS Biology, provided you address any remaining formatting and reporting issues. These will be detailed in an email that will follow this letter and that you will usually receive within 2-3 business days, during which time no action is required from you. Please note that we will not be able to formally accept your manuscript and schedule it for publication until you have made the required changes.

PRESS: We frequently collaborate with press offices. If your institution or institutions have a press office, please notify them about your upcoming paper at this point, to enable them to help maximise its impact. If the press office is planning to promote your findings, we would be grateful if they could coordinate with biologypress@plos.org. If you have not yet opted out of the early version process, we ask that you notify us immediately of any press plans so that we may do so on your behalf.

Sincerely,

Roli Roberts

Roland G Roberts, PhD 

Senior Editor 

PLOS Biology

rroberts@plos.org